

# The Largely Linear Response of Earth's Ice Volume to Orbital Forcing

Liam Wheen[1], Oscar Benjamin[1], Thomas Gernon[2,*], Cameron Hall[1,*], and Jerry Wright[1,*]

[1]Engineering Mathematics, University of Bristol, Queens Rd, Bristol, BS8 1QU, England
[2]Southampton Marine and Maritime Institute, University of Southampton, University Road, Southampton, SO17 1BJ, England
[*]These authors contributed equally to this work.

**Correspondence:** Liam Wheen (liam.wheen@bristol.ac.uk) and Oscar Benjamin (oscar.benjamin@bristol.ac.uk)

**Abstract.** Orbital forcing plays a key role in pacing the glacial-interglacial cycles. However, the mechanistic linkages between the orbital parameters — eccentricity, obliquity, and precession — and global ice volume remain unclear. Here, we investigate the effect of Earth's orbitally governed incoming solar radiation (that is, insolation) on global ice volume over the past 800,000 years. We consider a simple linear model of ice volume that imposes minimal assumptions about its dynamics. We find that
this model can adequately reproduce the observed ice volume variations for most of the past 800,000 years, with the notable exception of Marine Isotope Stage 11. This suggests that, aside from a few extrema, the ice volume dynamics primarily result from an approximately linear response to orbital forcing. We substantiate this finding by addressing some of the key criticisms of the orbitally forced hypothesis. In particular, we show that eccentricity can significantly vary the ocean temperature without the need for amplification on Earth. We also present a feasible mechanism to explain the absence of eccentricity's 400,000 year
period in the ice volume data. This requires part of the forcing from eccentricity to be lagged via a slow-responding mechanism, resulting in a signal that closer approximates the change in eccentricity. A physical interpretation of our model is proposed, using bulk ocean and surface temperatures as intermediate mechanisms through which the orbital parameters affect ice volume. These show reasonable alignment with their relevant proxy data, though we acknowledge that these variables likely represent a combination of mechanisms.

## 1 Introduction

For the past 800 thousand years (kyr), Earth's global ice volume has varied with a dominant period of approximately 100 kyr. These oscillations are referred to as the glacial-interglacial cycles and are demonstrated in Fig. 1. This figure also shows the mid-Pleistocene transition (MPT), which took place from approximately 1250 to 700 thousand years ago (kya) (Clark et al., 2006). This marks a significant change in ice volume dynamics whereby the dominant period shifted from 41 kyr to 100 kyr.

There is a clear link between Earth's orbital parameters and ice volume dynamics, as demonstrated by the power spectra in Fig. 2. Each of the prominent frequencies of ice volume appear to align with those of the orbital parameters. It is well understood that Earth's orbital configuration varies the magnitude and distribution of incoming solar radiation, known as insolation. However, the way in which this impacts Earth's cryosphere is still unclear. Most hypotheses can be categorised into two





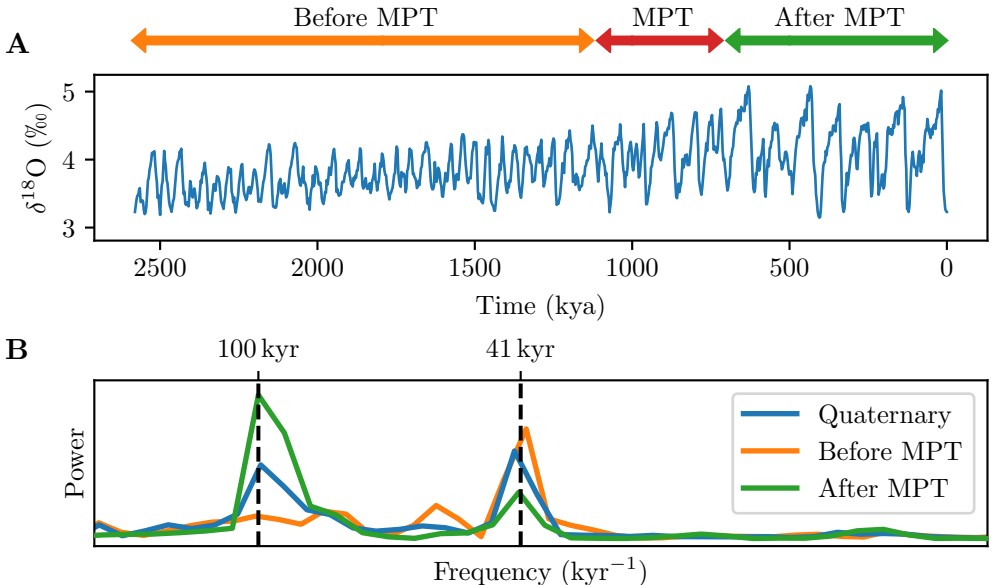

**Figure 1.** Time series (**A**) and power spectra (**B**) for the benthic $\delta^{18}$O stack from Lisiecki and Raymo covering the Quaternary period (Lisiecki and Raymo, 2005). The benthic $\delta^{18}$O ratio is commonly used as a proxy for global ice volume. The MPT spans approximately 550 kyr and marks a distinct change in dominant period, from 41 kyr to 100 kyr, as shown in the power spectra.

schools of thought. The first attributes the glacial-interglacial cycles to Earth-based mechanisms, with the orbital parameters
entraining the free oscillation. We refer to this as the *Intrinsic Forcing with Orbital Entrainment* (IFOE) approach. Examples
of this intrinsic forcing are interactions between the ice sheet and bedrock (Birchfield and Ghil, 1993b), $CO_2$ variation from
geological sources and sinks (Saltzman and Maasch, 1991; Paillard and Parrenin, 2004), and dust impacting Earth's surface
albedo (Peltier and Marshall, 1995). The opposing school of thought attributes the glacial-interglacial cycles to orbitally governed variations in insolation, with the potential for feedback mechanisms on Earth to amplify the effect. We refer to this as
the *Orbital Forcing with Potential Amplification* (OFPA) approach. In this paper, we present a model that provides evidence
for OFPA, as well as attempting to address some of the key limitations of the approach.

### 1.1 Background

Croll first proposed that the Earth's orbit influenced the glacial-interglacial cycles through insolation in the late 19th century (Croll, 1864). However, they incorrectly concluded that the ice sheets vary asynchronously across the hemispheres. With
the advent of more reliable data, Milankovitch was able to show that the glacial periods occur simultaneously around the globe,
with ice volume variations being more pronounced in the northern hemisphere (Milankovitch, 1930).

Since then, a number of IFOE and OFPA models have been developed to simulate ice volume in response to the orbitally
governed insolation. A common measure used to represent this forcing is the average daily insolation on the summer solstice



at $65°$ north, known as $Q_{65}$. In 1980, Imbrie and Imbrie modelled the change in ice volume as proportional to $Q_{65}$, supporting

the OFPA school of thought (Imbrie and Imbrie, 1980). Realising that the rate of ice growth is slower than its recession, they
included a conditional time constant that switches to capture the two rates.

In 2004, Paillard and Parrenin proposed a more complex system that was able to reproduce the MPT with a sliding param-
eter (Paillard and Parrenin, 2004). This model also included $Q_{65}$ as an input and a switching mechanism that depended on the
direction of ice volume change. They also included a variable that represents deep-water stratification, which depends on ice

volume induced changes in salinity. This feeds into a variable for atmospheric $CO_2$ which, in turn, affects ice volume, allowing
the model to produce unforced oscillations. This makes the model aligned with IFOE school of thought.

A later paper from Imbrie (2011) focussed on the interplay between the orbital parameters and how this affects ice volume,
again changing the dynamics depending on the direction in which the ice is changing (Imbrie et al., 2011). Though this
supports the OFPA approach, it can also produce free oscillations, meaning intrinsic dynamics could be captured by the model.

An important result of this paper is that it was able to recreate the MPT without using a time dependent parameter. This model
produces the closest fit of the three models discussed so far, but is very sensitive to parameter perturbations.

In the same year, Crucifix developed a Van der Pol style model that produces free oscillations with a 100 kyr period, which
is entrained by the $Q_{65}$ signal (Crucifix, 2011). This model aligns with the IFOE approach, though the main motivation was
to demonstrate that the ice volume dynamics are easily produced by a carefully tuned model. However, if the model is highly

sensitive to parameter perturbations, its predictions may be unreliable.

A common trait amongst these models is the assumption of a switching mechanism that depends on whether the ice volume
is growing or ablating. This is reasonable, given the sawtooth nature of the ice volume data in places (Fig. 1A), however,
there is no consensus on what this mechanism is. The inclusion of a switching mechanism also means that the model becomes
non-linear and highly sensitive to the choice of switching condition.

**1.2 Outline**

In this paper, we show that ice volume dynamics can mostly be explained by a linear OFPA model without the use of a switching
mechanism. Although our model supports the approach, OFPA has two commonly citepd limitations. Firstly, eccentricity is the
only parameter that oscillates with a period of around 100 kyr. However, as we show in Sec. 2, it can only vary the magnitude
of annual insolation by 0.2%. This has led some to argue that an Earth based amplification mechanism is necessary in order to

explain the 100 kyr period (Imbrie et al., 1993; Saltzman, 2001; Ganopolski and Calov, 2011). Another limitation of the OFPA
approach relates to the second prominent frequency of eccentricity. As shown in Fig. 2, the eccentricity signal also contains a
period of 400 kyr. This is not clearly discernible in the ice volume data, leading some studies to suggest that the 100 kyr period
instead results from the interplay of the other obliquity and precession (Liu et al., 2008; Saltzman et al., 1984; Ruddiman,
2003).

In this paper, we aim to address both of the previous issues, and in doing so, provide support for the OFPA school of thought.
We first address the notion that amplification of the eccentricity signal is necessary to explain its prominence in the ice volume
data. Using a simple model of bulk ocean temperature, we estimate the range of warming rates that eccentricity can induce.



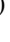

**Figure 2.** Time series (**A**) and power spectra (**B**) for the three orbital parameters and ice volume data, the sources for which are discussed in Appendices A and B. The ice volume power is logarithmically scaled to highlight the smaller peaks that align with precession. The dashed lines in **B** show all orbital frequencies aligning with frequencies in the ice volume data except for the 400.1 kyr peak in eccentricity.





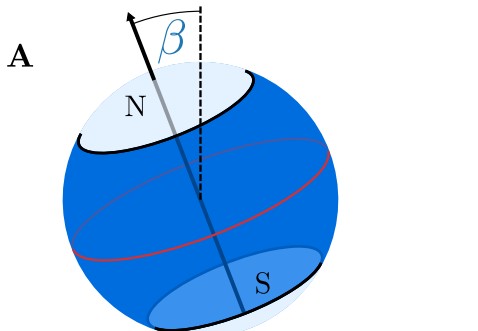

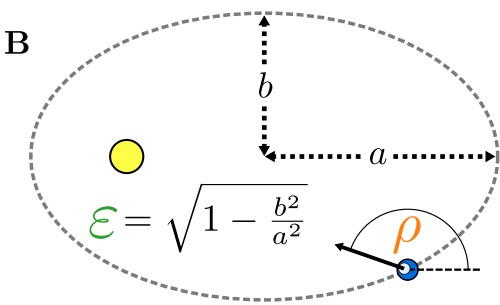

**Figure 3.** Diagrams of Earth defining the three orbital parameters. **A**: obliquity $\beta$ describes the tilt of Earth's rotational axis from vertical in the ecliptic frame. **B**: Eccentricity $\varepsilon$ is a function of the semi-major and semi-minor axes of Earth's orbit and has been exaggerated here for the purpose of visualisation. Precession $\rho$ describes the angle of Earth's rotational axis around the vertical axis in the ecliptic frame, measured from the aphelion.

Even with conservative parameter estimates, we find that the difference in rates is sufficient to explain the magnitude of ocean temperature dynamics, without the need for amplification.

We also propose that the 400 kyr period of eccentricity is absent from the ice volume data because, instead of its magnitude, it is the change in eccentricity that forces the ice volume dynamics. We find that the change in eccentricity contains a far weaker 400 kyr period and could arise from two terrestrial mechanisms that respond to eccentricity with different lags. We propose these could be the slow changing ocean temperature and the fast changing surface air temperature. If these mechanisms have opposing effects on ice volume, they can combine to produce a signal that resembles the change in eccentricity over time.

These findings are used to develop a phenomenological model of ice volume, comprising three instantaneous orbital parameter terms and a lagged eccentricity term. To evaluate the importance of the model parameters, we systematically prune them and refit the model whilst evaluating the performance each time. The key finding from this is that both the lagged and instantaneous eccentricity terms must be included in order to produce 100 kyr cycles with a consistent amplitude. Excluding either of these terms from the full model reduces the accuracy by approximately 40%, as is shown in Fig. 6.

We propose a physical interpretation for the components of our phenomenological model, whereby ocean temperature produces the lagged eccentricity signal and the 3 instantaneous orbital parameters describe the surface air temperature. Using our already fit parameter values, along with some extra constraints from data, we are able to model the dynamics of ice volume, ocean temperature, and surface air temperature. Each variable is compared with the equivalent proxy data over the past 800 kyr, showing reasonable agreement for most of the time period, supporting the validity of these intermediate physical variables.

## 2    Orbital Parameters


Our model uses Earth's three orbital parameters as inputs, the source for which is discussed in Appendix A. These are obliquity $\beta$, precession $\rho$, and eccentricity $\varepsilon$, which are shown in Fig. 3. As obliquity increases, it redistributes insolation away from the





equator towards the poles. Precession is the angle of Earth's rotational axis projected in the ecliptic plane and causes a larger
seasonal difference in insolation for one hemisphere, whilst reducing the difference in the other. To better reflect the sinusoidal
impact that this angle has on insolation, we use the cosine of precession in our model. Unlike obliquity and precession,
eccentricity changes the total insolation received by Earth over a year.

Glaciers spread down from the poles over land masses, and therefore exhibit more variation in the land-dense northern
hemisphere. Due to this asymmetry, it is logical that the redistribution of insolation from obliquity and precession would be
able to impact ice volume variation, despite not changing the total amount of insolation. Evidence of this impact is shown by
the power spectra in Fig. 2. As obliquity increases, it exposes the polar ice to more insolation, so we expect ice volume to
be inversely related to obliquity. Since we are measuring precession from the aphelion of Earth's orbit, the maximum of its
cosine coincides with maximum intensity insolation during the northern hemisphere summer. Milankovitch proposed that $Q_{65}$
correlates with glacial retreat, which would mean ice volume is inversely related to the cosine of precession (Milankovitch,
1930).

To understand how eccentricity affects insolation, we use the derivation from McGehee and Lehman (McGehee and Lehman,
2012). This gives the annually averaged insolation over Earth's surface as

$$\overline{Q_{\mathrm{E}}}(\varepsilon(t)) = \frac{K}{16\pi a^2 \sqrt{1 - \varepsilon(t)^2}}, \tag{1}$$

where $\varepsilon(t)$ is eccentricity over time, shown in Fig. 2, $a$ is the semi-major axis of Earth's orbit, and $K$ is the solar constant, both
of which are given in Table 1. Note that because this is annually averaged, it is independent of both obliquity and precession.

The Earth's orbital eccentricity varies between approximately 0 and 0.06, as shown in Fig. 2. If we substitute these limits into
$\overline{Q_{\mathrm{E}}}$, along with the constants given in Table 1, we find the annually averaged insolation ranges from 340.353 to 340.967 W/m$^2$.
This corresponds to a maximum change of 0.18% due to eccentricity.

This seemingly insignificant variation has led some researchers to propose the need for an Earth-based amplifier in order for
eccentricity to have any significant impact on ice volume (Imbrie et al., 1993; Saltzman, 2001; Ganopolski and Calov, 2011).
Although we do not rule out the existence of these amplifiers, it is also true that the 100 kyr period of eccentricity allows for a
prolonged increase or decrease in insolation to span many thousands of years. This could potentially produce gradual changes
that build up in the Earth system, resulting in a significant impact on ice volume. We propose that a mechanism that is slow to
equilibrate, such as ocean temperature, could partially store the excess energy produced as eccentricity increases, allowing for
significant warming to occur over thousands of years.

We propose that eccentricity is capable of explaining the broad dynamics of ocean temperature, as observed in the paleocli-
mate record shown in Fig. 9. In order to test this hypothesis, we estimate how eccentricity affects the rate of change of ocean
temperature and evaluate if there is enough energy variation in the system to produce the observed range in ocean temperature.

In this derivation, we exclude the surface ocean since its interface with the atmosphere causes it to behave differently than
the rest of the ocean. We will instead consider the bulk ocean temperature $O(t)$, relating to all water below a certain depth. We
also treat the heat loss rate of the bulk ocean as constant over time, however we will improve upon this assumption in Sec. 4.5.





The rate of change of the bulk ocean temperature is given by

$$\frac{\mathrm{d}O}{\mathrm{d}t} = \frac{\text{Total Absorbed Power}}{\text{Ocean Heat Capacity}} - \text{Heat Loss Rate}. \tag{2}$$

The total power provided to the bulk ocean through insolation can be expressed as

$$P_O(\varepsilon(t)) = \alpha\gamma A_O \overline{Q_{\mathrm{E}}}(\varepsilon(t)), \tag{3}$$

which has units W. The $\alpha$ coefficient accounts for the absorption and reflection that occurs whilst passing through the atmosphere. The coefficient $0 < \gamma \leq 1$ accounts for the heat emitted back to the atmosphere from the surface ocean layer before it can reach the bulk volume of the ocean, where $\gamma = 1$ implies no loss of heat to the atmosphere. The annually averaged insolation reaching Earth $\overline{Q_{\mathrm{E}}}$ comes from (1), which is then scaled by the total area of the ocean $A_O$. For this approximate calculation we are ignoring the latitudinal asymmetry of ocean water, treating it as uniformly distributed across Earth's surface. All of these values, apart from the free parameter $\gamma$, are given in Table 1.

This power is then divided by the total heat capacity of the ocean, given as

$$C_O = \rho_O V_O c_O, \tag{4}$$

which has units J/°C. The mass of the ocean is given by $\rho_O V_O$, where $\rho_O$ is the average density of the ocean and $V_O$ is the total volume. Although heat capacity varies with the salinity and temperature of the ocean, we will express the average specific heat capacity with $c_O$.

We can now substitute in our measured values to estimate the bulk ocean warming rate. Since this will have units of °C/s, we rescale by $3.1536 \times 10^{10}$ to attain the rate in °C/kyr.

This gives us a warming rate of

$$\frac{\mathrm{d}O}{\mathrm{d}t} = \frac{P_O(\varepsilon(t))}{C_O} - l = \frac{353.907\gamma}{\sqrt{1 - \varepsilon(t)^2}} - l\,°\text{C/kyr}, \tag{5}$$

where $l$ is the constant heat loss term and $\gamma$ is the absorption constant. Since we have not formally defined the depth at which our bulk ocean meets the ocean surface, we do not have a value for $\gamma$. Instead, we treat this, and $l$ as free parameters to be determined through fitting. If eccentricity is capable of producing the observed ocean temperature changes without amplification, then we would expect a fit value of $\gamma < 1$. A fit value of $\gamma > 1$ is not physically valid and would suggest that there is some amplification mechanism that we have not accounted for.

As a simple calculation, we explore the role $\gamma$ plays in this warming rate, using ocean temperature data as a reference. The last glacial maximum occurred around 20 kya, and since then our ocean has heated at an extreme rate, similar to other transitions into an interglacial period (Clark et al., 2009). The surface water temperature (SWT) has increased by up to 10°C in individual locations (Pahnke and Sachs, 2006). However, globally averaged SWT and bottom water temperature (BWT) have increased by around 3 to 5°C respectively (Elderfield et al., 2012; Annan and Hargreaves, 2013). This gives us an approximate warming rate of 0.2°C/kyr.



**Table 1.** Constants used in this paper.

| Term | Symbol | Value | Units | Source |
|---|---|---|---|---|
| Solar Luminosity | $K$ | $3.8287 \times 10^{26}$ | W | (Kopp and Lean, 2011) |
| Semi-Major Axis | $a$ | $1.4960 \times 10^{11}$ | m | (Kopp and Lean, 2011) |
| Ocean Surface Absorption Ratio | $\alpha$ | 0.48 | 1 | (Faizal and Rafiuddin Ahmed, 2011) |
| Average Ocean Density | $\rho_O$ | 1025 | kg/m$^3$ | (Talley, 2011) |
| Ocean Specific Heat | $c_O$ | 3850 | J/kg $^\circ$C | (Talley, 2011) |
| Ocean Volume | $V_O$ | $1.335 \times 10^{18}$ | m$^3$ | (Eakins and Sharman, 2007) |
| Ocean Surface Area | $A_O$ | $3.619 \times 10^{14}$ | m$^2$ | (Eakins and Sharman, 2007) |
| Ocean Heat Capacity | $C_O$ | $5.268 \times 10^{24}$ | J/$^\circ$C | (Talley, 2011; Eakins and Sharman, 2007) |
| Average Eccentricity | $\overline{\varepsilon}$ | 0.02707 | 1 | (Laskar et al., 2004) |
| Average Obliquity | $\overline{\beta}$ | 0.40739 | rad | (Laskar et al., 2004) |
| Average Ocean Input Power | $P_O(\overline{\varepsilon})$ | $5.914 \times 10^{16}$ | J/s | (Laskar et al., 2004; Faizal and Rafiuddin Ahmed, 2011; Kopp and Lean, 2011; Eakins and Sharman, 2007) |

If we suppose that half of the incoming heat reaches the bulk ocean, meaning that $\gamma = 0.5$, then for an eccentricity range of $0 \leq \varepsilon \leq 0.06$, (5) would give a difference in warming rates of $0.319^\circ$C/kyr. In Sec. 4.5, we find from fitting that $\gamma = 0.59$, resulting in a similar rate as calculated here, and also of the same scale as the $0.2^\circ$C/kyr we see in the data. This suggests that eccentricity could explain the observed ocean temperature changes without the need for amplification.

It is important to note that we have used the extrema of eccentricity and ocean temperature for this calculation. We also acknowledge that the ocean heat loss rate will vary as a function of temperature to some degree. However, this simple calculation shows that, given the range for eccentricity, it has the ability to drive a change in ocean temperature on the same scale as that observed in the data.

## 3   Phenomenological Model

We now present a simple phenomenological model that is intended to reproduce the global ice volume data using only a linear function of the orbital parameters. We then fit the coefficients of this model to the data and evaluate its predictive power. Although we do not expect this model to outperform the more complex, non-linear ice volume models, it will demonstrate the extent to which a linear OFPA model can approximate the ice volume dynamics. We justify the inclusion of each term in our model by refitting it with every subset of terms, evaluating the degree to which it can explain the data in each case.

The model is based on the assumption that the change in ice volume depends instantaneously on the three orbital parameters, as well as a lagged version of eccentricity. The instantaneous terms allow us to reproduce the orbitally aligned power spikes in the ice volume data from Fig. 2. The lagged term interacts with the instantaneous eccentricity term of the opposite sign to





approximate the change in eccentricity. This effectively removes the 400 kyr period from the ice volume solution to better align with the data, as shown in Fig. 7.

To achieve this, the model comprises two linear differential equations, describing the evolution of ice volume $I(t)$, and the slow-responding variable $\tilde{\varepsilon}(t)$, which resembles the eccentricity signal with a lag of approximately $\tau$. This is because $\tilde{\varepsilon}$ changes according to the difference between itself and eccentricity $\varepsilon$, converging towards $\varepsilon$ at a rate determined by $\tau$. If $\tau = 0$ then (6) would simplify to $\tilde{\varepsilon}(t) = \varepsilon(t)$, meaning there is no lag between the two signals. However, as $\tau$ increases, $\tilde{\varepsilon}$ converges more slowly towards $\varepsilon$, and so the lag between the two signals increases. Although we have suggested that ocean temperature

could be this slow-responding variable, here we are purely concerned with reproducing the ice volume data with no physical interpretation or additional assumptions.

The model is therefore given by

$$\tau \frac{\mathrm{d}\tilde{\varepsilon}}{\mathrm{d}t} = \varepsilon(t) - \tilde{\varepsilon}(t), \tag{6}$$

$$\tau \frac{\mathrm{d}I}{\mathrm{d}t} = p_1 \tilde{\varepsilon}(t) - p_2 \varepsilon(t) - p_3 \beta(t) - p_4 \cos(\rho(t)) - I(t) + p_5, \tag{7}$$

where the $p_i$ coefficients are to be fitted, along with the time constant $\tau$. This is introduced since we do not expect ice volume, nor the slow variable, to respond instantaneously to changes in the orbital parameters. Two separate time constants were originally used for this model, however, fitting revealed the two constants to be approximately equal. This could be due to some coupling between ice volume and the mechanism that $\tilde{\varepsilon}(t)$ represents, causing them to change at the same rate.

We now present the analytical solution to this model and fit the coefficients to the ice volume data. Using the integrating

factor method, we solve for the slow variable to get

$$\tilde{\varepsilon}(t) = \zeta_\tau[\varepsilon(t)] + \tilde{\varepsilon}_0 e^{-t/\tau}, \tag{8}$$

where $\tilde{\varepsilon}_0$ is the initial condition and the functional

$$\zeta_\tau[y(t)] = \frac{e^{-t/\tau}}{\tau} \int_0^t y(u) e^{u/\tau} \mathrm{d}u, \tag{9}$$

for some function $y(t)$.

Substituting this into the differential equation for ice volume gives

$$\tau \frac{\mathrm{d}I}{\mathrm{d}t} = p_1 \zeta_\tau[\varepsilon(t)] - p_2 \varepsilon(t) - p_3 \beta(t) - p_4 \cos(\rho(t)) + p_5 + p_1 \tilde{\varepsilon}_0 e^{-t/\tau}. \tag{10}$$

We then solve this using the integrating factor method again to get

$$I(t) = p_1 \zeta_\tau\left[\zeta_\tau[\varepsilon(t)]\right] - p_2 \zeta_\tau[\varepsilon(t)] - p_3 \zeta_\tau[\beta(t)] - p_4 \zeta_\tau[\cos(\rho(t))] + p_5 + \left(\frac{p_1 \tilde{\varepsilon}_0 t}{\tau} + I_0 - p_5\right) e^{-t/\tau}. \tag{11}$$

If we run our model for sufficiently long before our period of interest, the $e^{-t/\tau}$ term in $I(t)$ can be treated as zero. This

gives the asymptotic approximation as

$$I(t) = p_1 \zeta_\tau\left[\zeta_\tau[\varepsilon(t)]\right] - p_2 \zeta_\tau[\varepsilon(t)] - p_3 \zeta_\tau[\beta(t)] - p_4 \zeta_\tau[\cos(\rho(t))] + p_5. \tag{12}$$





**Table 2.** Parameter values for the phenomenological model given in (7). Their roles are shown in Fig. 6B. Errors given by the 95% confidence interval in the fit are shown beside the estimated value.

| Parameter | Role | Value | Units |
|:---:|:---:|:---:|:---:|
| $p_1$ | Slow Eccentricity | $1.88 \pm 0.05$ | $\times 10^9 \text{ km}^3$ |
| $p_2$ | Fast Eccentricity | $1.94 \pm 0.05$ | $\times 10^9 \text{ km}^3$ |
| $p_3$ | Fast Obliquity | $1.54 \pm 0.06$ | $\times 10^9 \text{ km}^3$ |
| $p_4$ | Fast Precession | $1.9 \pm 0.1$ | $\times 10^7 \text{ km}^3$ |
| $p_5$ | Constant Offset | $6.8 \pm 0.2$ | $\times 10^8 \text{ km}^3$ |
| $\tau$ | Response Rate | $14.8 \pm 0.4$ | kyr |

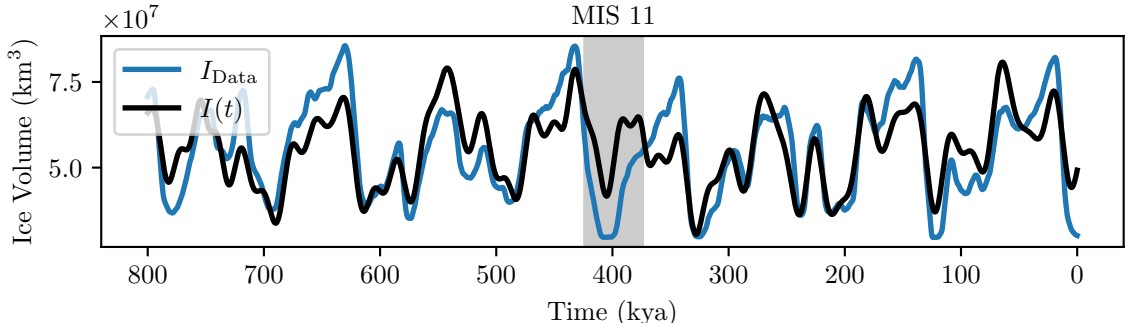

**Figure 4.** Our modelled ice volume $I(t)$ from (12), alongside the ice volume data $I_{\text{Data}}$. The grey region delineates Marine Isotope Stage (MIS) 11, around which there is a notable difference between the two curves. The model parameters that produce this fit are given in Table 2.

This solution is non-linear in $\tau$, so to optimise the parameters of the model, we repeat a least squares fit for the $p_i$ coefficients whilst varying $\tau$, guaranteeing that the parameters are globally optimised. The optimal parameter values for this model are given in Table 2, with the corresponding solution shown in Fig. 4.

The ice volume solution in Fig. 4 shows reasonably close agreement with the data, with the notable exception of the time interval around Marine Isotope Stage (MIS) 11, which will be discussed in Sec. 5. We emphasise that this model is not intended to be a perfect fit to the data, but rather a demonstration of how much can be explained by a linear dependence on the orbital parameters. By reproducing the majority of the ice volume data with this linear model, we propose that more complex mechanisms are only needed to explain the few remaining extrema such as MIS 11.

The analytical solution for ice volume is expressed in terms of the $\zeta_\tau$ functional. To demonstrate how $\zeta_\tau$ responds to its inputs, Fig. 5 shows each term in (12) plotted alongside the orbital parameter it depends on. We see that the effect of $\zeta_\tau$ is somewhat different in each case. For eccentricity, a lag approximately equal to $\tau = 14.8 \text{ kyr}$ is introduced each time $\zeta_\tau$ is applied. For obliquity and precession, the amplitude is reduced more significantly, and the lag is shorter. This is because they oscillate at a higher frequency than eccentricity. As a result, where the $\zeta_\tau[\varepsilon(t)]$ curve is able to slowly follow the $\varepsilon(t)$ curve,



$\zeta_\tau[\beta(t)]$ and $\zeta_\tau[\cos(\rho(t))]$ cannot reach the extrema of their inputs before they begin to change direction. However, since the scale factors in (12) will account for this effect, the only important feature of $\zeta_\tau$ is the lag it introduces. The ice volume solution can therefore be interpreted simply as a weighted sum of the lagged orbital parameters, where eccentricity appears twice, with a longer lag the second time.

      In order to evaluate the necessity of each term in our model, we systematically prune terms whilst evaluating the accuracy
in each case. The results of this are shown in Fig. 6. To measure the accuracy, we are using the variance of the ice volume data that is explained by the model. The offset term $p_5$ and time constant $\tau$ are included in every version of the model so that the solutions are comparable. The precession term, represented by $p_4$, is shown to consistently contribute less to the variance explained than the other terms. This is partly due to the nature of the ice volume curve, in which the higher frequencies appear with smaller amplitudes. However, this may also arise due to solid Earth forcing, such as volcanic activity, that could affect the
ice volume on this timescale, introducing noise that precession cannot account for. Although precession contributes the least, it is still responsible for approximately 5% of the variance in the ice volume data, regardless of the other terms included in the model. This suggests that it is not contributing to over-fitting. Moreover, its inclusion allows for the higher frequencies, shown in Fig. 1, to be represented in the model.

      Secondly, we find that the subsets that include both $p_1$ and $p_2$, relating to the slow and fast eccentricity terms, consistently
produce the best fit. This indicates that the change in eccentricity represented by the pair is crucial to the model, more so than either term on its own. In order to better understand the role of the slow variable $\tilde{\varepsilon}(t)$, we examine how eccentricity relates to ice volume. As shown in Fig. 2, ice volume decreases during peaks of eccentricity, which is consistent with (1). However, the degree to which ice volume decreases appears to be independent of the magnitude of eccentricity's peaks. This suggests that the absolute value of eccentricity may be less important than the direction in which it is changing. By including the $\tilde{\varepsilon}(t)$
variable alongside $\varepsilon(t)$, we are providing the model with both current, and lagged, values of eccentricity. The difference of these two signals represents the change in eccentricity over time, shown by the first two terms in (10).

      As shown in Table 2, the fit $p_1$ and $p_2$ values are within 3% of each other. This indicates that the optimised model only depends on the change in eccentricity and not the instantaneous or lagged values on their own. We could therefore simplify the model by setting $p_2 = p_1$, without losing accuracy. However, since we wish to find physical interpretations for both the slow
and fast responses to eccentricity we will keep the two terms distinct.

      By using the relatively short-term change in eccentricity as input, rather than eccentricity itself, our model is able to produce 100 kyr oscillations whilst effectively removing the 400 kyr amplitude modulation. Fig. 7A shows the prominent 400 kyr peak in eccentricity's power spectrum that is significantly reduced in the eccentricity component of our model solution $I_\varepsilon(t)$. Fig. 7B shows this effect in the time domain, with $I_\varepsilon(t)$ more closely matching the broad dynamics of the ice volume data.



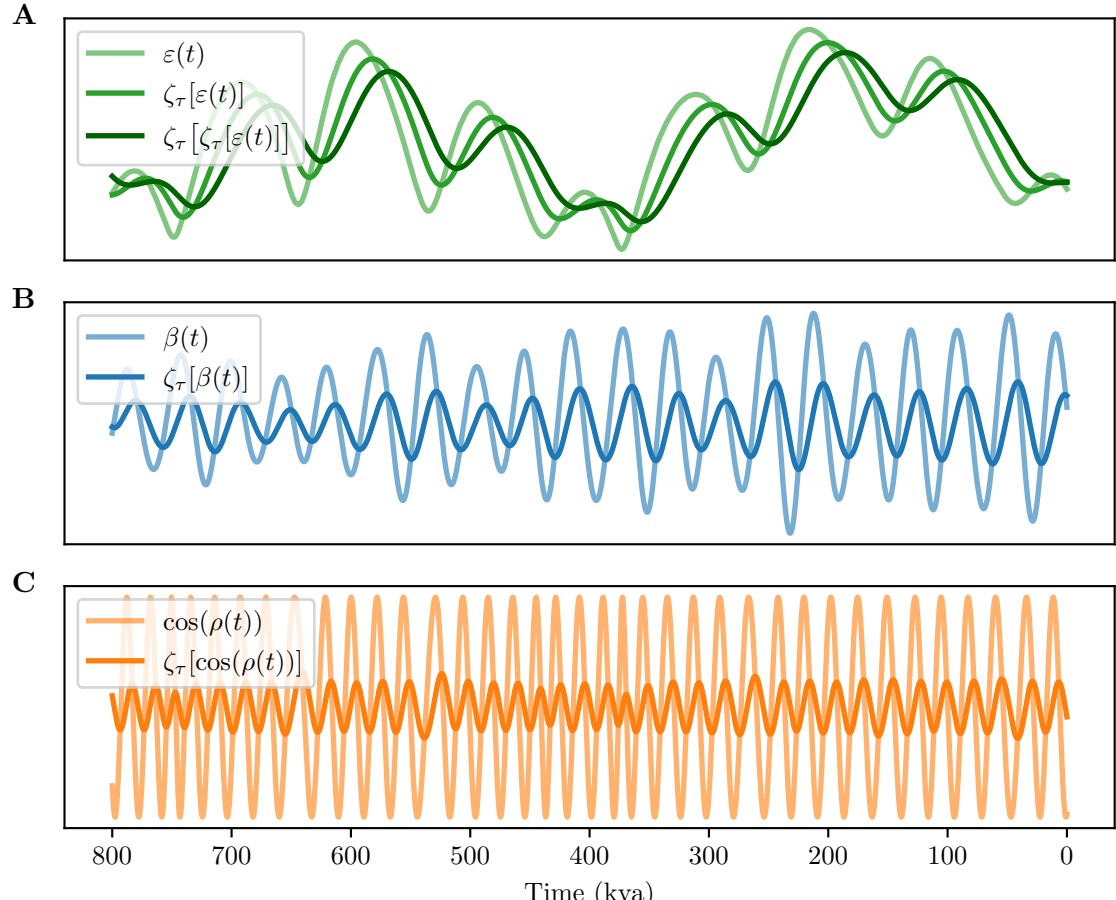

**Figure 5.** The qualitative effect of the functional $\zeta_\tau$ on eccentricity (**A**), obliquity (**B**), and the cosine of precession (**C**) as defined by (9), where $\tau = 14.8\,\text{kyr}$. Each $\zeta_\tau$ term that appears in (12) is shown. Note how the higher the frequency of the orbital parameter, the more $\zeta_\tau$ reduces its amplitude.





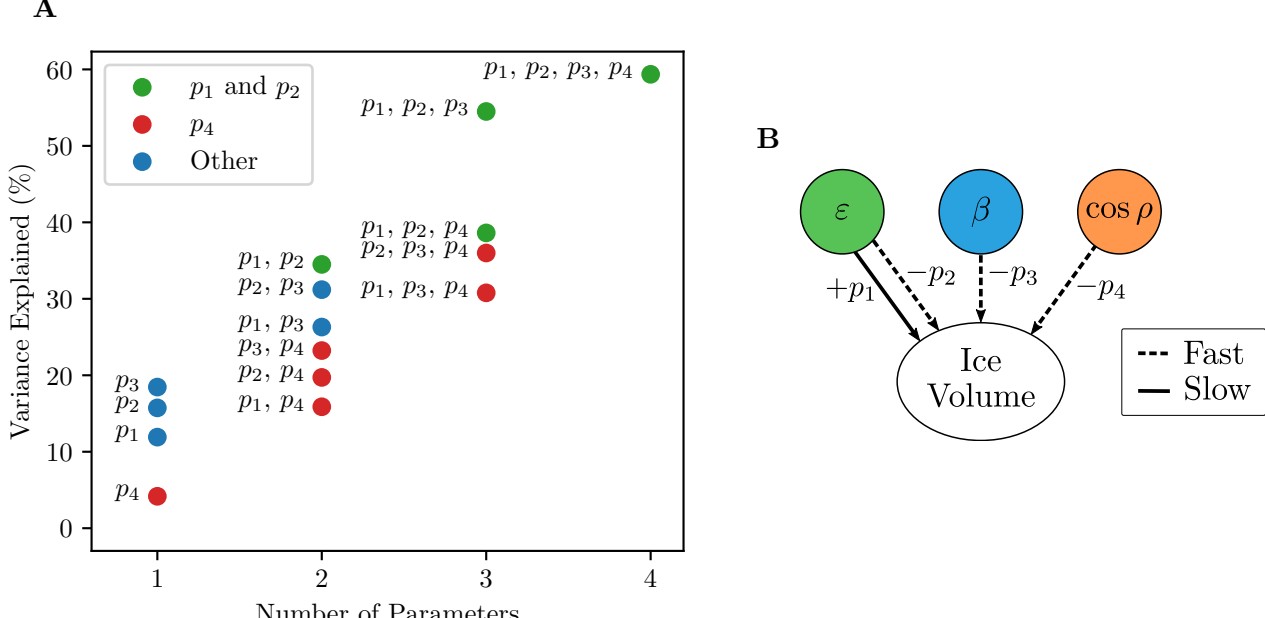

**Figure 6. A**: Variance explained for all possible parameter combinations of the model given by (7), with the excluded parameters set to zero. The constant term $p_5$ and time constant $\tau$ are always included. For each case, the included parameters were optimised to attain the best fit to the ice volume data. Cases where both $p_1$ and $p_2$ are included (green) produce especially good fits, whilst cases without this pair, but with $p_4$ included (red), produce especially poor fits. **B**: Flow diagram showing the role of each parameter in the model. The fast arrows (dashed) represent an instantaneous response, scaled by the accompanying parameter, whilst the slow arrows (solid) depict a response to the input that is scaled by the accompanying parameter and lagged by approximately $\tau$.

## 4 Physical Interpretation

So far, we have presented a phenomenological model that is able to capture the broad dynamics of the ice volume data. We now propose a physical interpretation for the terms of this model and attempt to produce solutions for these physical mechanisms that are consistent with the relevant data.

### 4.1 Choice of Mechanisms

The phenomenological model contains three terms that respond quickly to the orbital parameters, and one that responds slowly. We propose that these slow and fast dynamics are attributed to separate mechanisms. The fast part of the model could represent the response of Earth's surface air temperature (SAT) to the orbitally varied insolation, which can be considered instantaneous on this timescale. As discussed in Sec. 2, an increase in eccentricity, obliquity, or precession can reduce ice volume. This predominantly occurs through SAT and direct radiation of the glaciers. As these are closely correlated effects, and proxy data





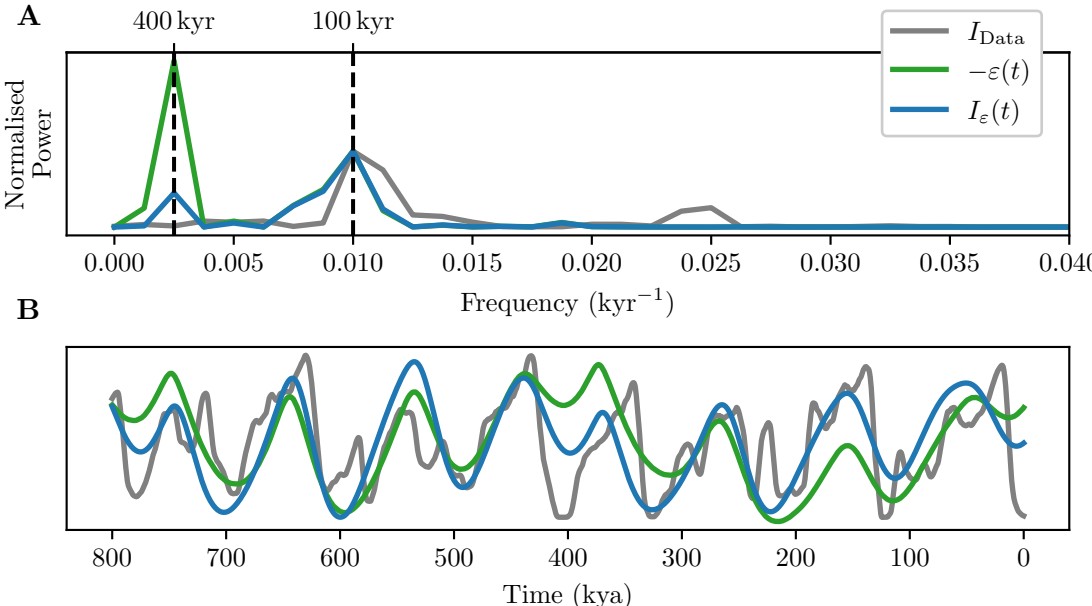

**Figure 7. A**: Power spectra comparison of the ice volume data, the negative eccentricity curve $-\varepsilon(t)$, and the eccentricity component of our model solution $I_\varepsilon(t)$. They have been normalised to equate the powers corresponding to 100 kyr. Our $I_\varepsilon(t)$ power spectrum matches that of $\varepsilon(t)$ apart from a significant drop around the 400 kyr period. **B**: Time series showing the same comparison, also normalised for qualitative comparison. They show how the filtered eccentricity better captures the broad behaviour of the ice volume curve, namely around 780, 400, 320, and 160 kya.

only exists for SAT, we treat this as a combined surface temperature variable $S(t)$ and use SAT data as a proxy for its qualitative behaviour.

We propose that the slow variable $\tilde{\varepsilon}(t)$ relates to ocean temperature $O(t)$ in some way. Due to its high thermal inertia, it could take thousands of years for the ocean to equilibrate to a new temperature. This response time is slow enough that the impact of precession and obliquity are significantly reduced, as can be seen in Fig. 5. To confirm this, we trialled an ocean

model with all three orbital parameters and found no significant improvement to the accuracy of the fit.

Although we are treating $O(t)$ as ocean temperature, it may also incorporate mechanisms such as long-term feedbacks in atmospheric $CO_2$. The ocean is one of the largest carbon sinks on Earth, but holds less $CO_2$ at higher temperatures (Kitzmann et al., 2015). This positive correlation between ocean temperature and atmospheric $CO_2$ concentration means it is difficult to separate the impact of each on ice volume. However, as with SAT and radiation, these effects are closely correlated, and so we

choose to represent these correlated mechanisms with our single slow variable $O(t)$. This allows us to use ocean temperature data as a proxy for its qualitative behaviour.



We are using the fit parameters from our phenomenological model, which means we are assuming ocean temperature has a time constant of $\tau = 14.8\,\text{kyr}$. However, since the temperature of the ocean never reaches a global equilibrium, it is difficult to determine if this is physically plausible. Van Aken states that heat can be mixed across all depths on a time scale of hundreds

to a few thousand years (Van Aken, 2007). The Intergovernmental Panel on Climate Change state that deep ocean temperatures can take thousands of years to respond to surface temperature changes (Stocker, 2014). Crucifix estimates an even longer mixing time on the order of $10\,\text{kyr}$ (Crucifix, 2011). There is a similar level of uncertainty when it comes to estimating the time constant for ice sheet growth and ablation. Estimates range from 500 years on the lower end (Ritz et al., 2001), but can go up to $27\,\text{kyr}$ (Oerlemans, 1980). Our fit time constant of $\tau = 14.8 \pm 0.4\,\text{kyr}$ therefore seems feasible for both ocean temperature

and ice volume.

Although we have demonstrated that the observed warming rate of the ocean can be sufficiently explained by eccentricity, we have so far treated the heat loss rate as constant. Heat loss occurs through back radiation, convection, conduction, and evaporation (Faizal and Rafiuddin Ahmed, 2011). We will assume for simplicity that these are all linearly dependent on bulk ocean temperature $O(t)$, giving a consolidated loss term of $mO(t) + n$. Replacing the constant loss rate in (5) with this linear

loss rate gives

$$\frac{\mathrm{d}O}{\mathrm{d}t} = \frac{353.907\gamma}{\sqrt{1 - \varepsilon(t)^2}} - mO(t) - n, \tag{13}$$

which has units °C/kyr.

As we wish to substitute $O(t)$ into our linear model, we take the first order Taylor expansion about the average eccentricity $\overline{\varepsilon}$ to get

$$\frac{\mathrm{d}O}{\mathrm{d}t} = 9.591\gamma\varepsilon(t) + 353.8\gamma - mO(t) - n, \tag{14}$$

where $\overline{\varepsilon}$ is given in Table 1.

In order to simplify this equation for use in our physical model, we write it as

$$\tau\frac{\mathrm{d}O}{\mathrm{d}t} = c\varepsilon(t) - O(t) + \alpha_O, \tag{15}$$

where

$$\tau = \frac{1}{m}, \tag{16}$$

$$c = \frac{9.591\gamma}{m}, \tag{17}$$

$$\alpha_O = \frac{353.8\gamma - n}{m}. \tag{18}$$

Once we have estimated values for $c$, and $\alpha_O$, we can estimate all of the coefficients in (14). For this equation to be physically plausible, we would expect the resultant heat gain and heat loss rates to be approximately equal, so as to avoid unbounded

temperature change. We also require that $0 < \gamma \leq 1$ is satisfied.



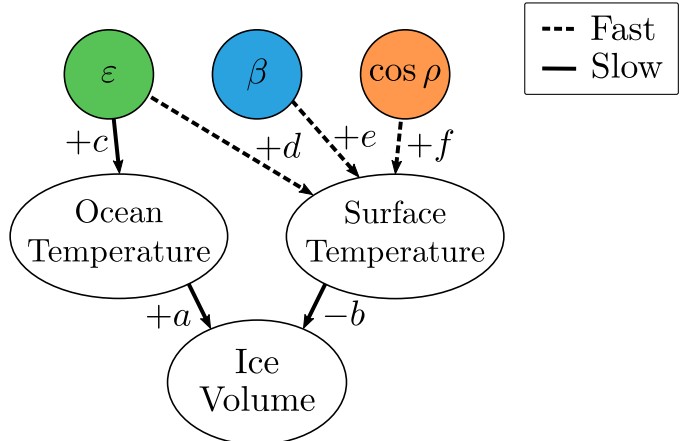

**Figure 8.** Flow diagram showing the proposed physical model of orbital influence through bulk ocean and surface air temperature. The fast arrows represent an instantaneous response, scaled by the accompanying parameter, whilst the slow arrows depict a response to the input that is scaled by the accompanying parameter and lagged by approximately $\tau$.

## 4.2 Physical Model

Our physical model will produce the same ice volume $I(t)$ solution as the phenomenological model, but will also produce solutions representing bulk ocean temperature $O(t)$ and SAT $S(t)$. These three variables can then be compared to their respective proxy data to help validate the model. To remain consistent with the phenomenological model, we require that ocean temperature positively impacts ice volume, whilst SAT negatively impacts it. It is logical that an increase in SAT would lead to a decrease in ice volume, however it is less obvious how an increase in ocean temperature could lead to an increase in ice volume.

A possible explanation for this relates to the evaporation rate from the ocean surface. As the ocean temperature increases, the evaporation rate also increases, leading to an increase in the moisture content of the air. This could then lead to greater precipitation over the ice sheets, increasing their volume. It is difficult to determine the degree to which this would impact ice volume, especially since ocean warming could also be attributed to a decrease in ice volume. However, we emphasise that $O(t)$ does not necessarily represent global ocean temperature, with one alternative being the ocean temperature only at high latitudes. In this case, it is possible that increased moisture in the atmosphere plays a more significant role.

The flow diagram in Fig. 8 shows this proposed physical model, extending the phenomenological model from Sec. 3. The governing equations for this model are

$$\tau \frac{\mathrm{d}I}{\mathrm{d}t} = aO(t) - bS(t) - I(t) + \alpha_I, \tag{19}$$

$$\tau \frac{\mathrm{d}O}{\mathrm{d}t} = c\varepsilon(t) - O(t) + \alpha_O, \tag{20}$$

$$S(t) = d\varepsilon(t) + e\beta(t) + f\cos(\rho(t)) + \alpha_S, \tag{21}$$





where $a$, $b$, $c$, $d$, $e$, $f$, $\alpha_O$, $\alpha_I$, $\alpha_S$, and $\tau$ are parameters that will be determined. In order to produce three distinct solutions that can be compared to their relevant data, each variable requires an offset and scaling factor. As a result, this physical model contains 10 parameters, compared to the 6 in the phenomenological model. In Sec. 4.5, we use the proxy data for ocean temperature and SAT to introduce 4 extra constraints, allowing us to uniquely determine the physical parameter values.

In order to analytically solve this system, we once again solve for the slow variable $O(t)$ using an integrating factor. This is then substituted into the ice volume equation, alongside the instantaneous $S(t)$. We then solve for $I(t)$ using an integrating factor. The solution for $O(t)$ is given as

$$O(t) = c\zeta_\tau[\varepsilon(t)] + \alpha_O - \alpha_O e^{-t/\tau} + O_0 e^{-t/\tau}, \tag{22}$$

where $O_0$ is the initial condition for $O(t)$ and the functional $\zeta_\tau$ is defined in (9).

Substituting (21) and (22) into (19) then gives

$$\tau\frac{\mathrm{d}I}{\mathrm{d}t} = ac\zeta_\tau[\varepsilon(t)] - bd\varepsilon(t) - be\beta(t) - bf\cos(\rho(t)) - I(t) + (a\alpha_O - b\alpha_S + \alpha_I) + a(O_0 - \alpha_O)e^{-t/\tau}. \tag{23}$$

The analytical solution for $I(t)$ is then

$$I(t) = ac\zeta_\tau\big[\zeta_\tau[\varepsilon(t)]\big] - bd\zeta_\tau[\varepsilon(t)] - be\zeta_\tau[\beta(t)] - bf\zeta_\tau[\cos(\rho(t))] \tag{24}$$

$$+ (a\alpha_O - b\alpha_S + \alpha_I) + \left(\frac{a}{\tau}\big((O_0 - \alpha_O)t - \alpha_O\tau\big) + b\alpha_S - \alpha_I + I_0\right)e^{-t/\tau}. \tag{25}$$

As before, if we solve from sufficiently long before our period of interest, we can asymptotically approximate this as

$$I(t) = ac\zeta_\tau\big[\zeta_\tau[\varepsilon(t)]\big] - bd\zeta_\tau[\varepsilon(t)] - be\zeta_\tau[\beta(t)] - bf\zeta_\tau[\cos(\rho(t))] + (a\alpha_O - b\alpha_S + \alpha_I), \tag{26}$$

$$= p_1\zeta_\tau\big[\zeta_\tau[\varepsilon(t)]\big] - p_2\zeta_\tau[\varepsilon(t)] - p_3\zeta_\tau[\beta(t)] - p_4\zeta_\tau[\cos(\rho(t))] + p_5, \tag{27}$$

where we have included the parameters from the phenomenological model from (12) to demonstrate how its coefficients align with those of the physical model. As is shown by this comparison, although the physical model has 10 parameters, the two solutions for ice volume are still equivalent. Where we originally had single $p_i$ coefficients scaling the orbital parameters, we now have a product of two coefficients, one scaling the intermediate physical variable and the other scaling its contribution to ice volume. Similarly, the $p_5$ coefficient is now represented by the weighted sum of three offsets, one for each physical variable. In order to understand the physical quantities that we are modelling, we review the available proxy data for each one and compare it to our model solutions.

## 4.3 Ocean Temperature

The ocean temperature time series shown in Fig. 9 shows both BWT and SWT proxy data. The blue and orange plots represent the global BWT and show reasonable agreement with each other. The blue plot comes from Elderfield, who used Mg/Ca ratios to separate the BWT and ice volume contributions to benthic foraminifera $\delta^{18}$O data (Elderfield et al., 2012). The orange plot comes from Shackleton's quadratic model of BWT from $\delta^{18}$O, fit using current samples of foraminifera for which the local



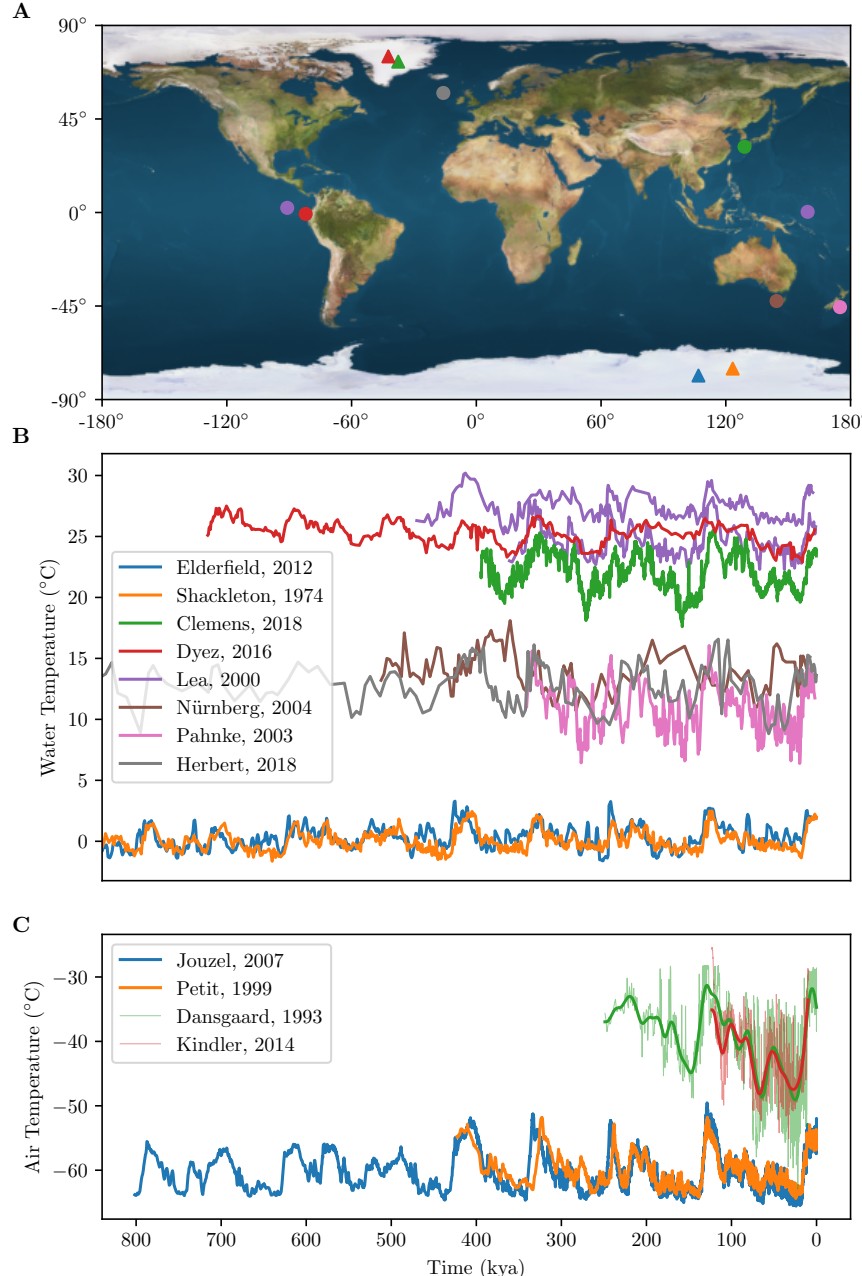

**Figure 9. A**: Drilled core locations used in our data comparison with seafloor drill sites shown as circles and ice core drill sites shown as triangles (ear). **B**: Proxy data for BWT and SWT from independent drilled cores, their colours correspond to the locations shown on the map. The global BWT data are plotted in blue and orange with the remaining plots relating to the regionally dependent SWT data. **C**: Regional SAT for locations in Antarctica and Greenland. Since the Greenland data are more noisy, a second order low-pass Butterworth filter was used to remove frequencies above $19 \, \text{kyr}^{-1}$, as this is the highest frequency at which orbital parameters vary.





BWT is known (Shackleton, 1974). This model was then applied to the Lisiecki and Raymo $\delta^{18}$O data, which largely comprises the same foraminifera used to calibrate Shackleton's model (Lisiecki and Raymo, 2005).

The remaining 7 plots relating to water temperature correspond to SWT proxy data, which vary far more with location. Data sources were chosen to incorporate drill sites from a range of latitudes, as shown on the accompanying map. The SWT data are produced through a method similar to that used by Elderfield, with the addition of a second measure that relies on alkenones. These are a type of ketone that some phytoplankton produce, with more saturated alkenones being produced in higher temperatures. Since phytoplankton live near the surface of the water to collect sunlight, the degree of alkenone saturation

found in these phytoplankton can act as a proxy for SWT (Ho et al., 2013).

### 4.4   Surface Air Temperature

The third variable in our physical model represents globally averaged SAT. Proxy data on the timescale we are interested in are limited, with the predominant source being continental ice cores (Fig. 9A). As snow accumulates, the top layers slowly compress lower layers into ice, preserving the isotopic composition of the snow at the time. The ice cores containing the longest

history are located in Antarctica and Greenland, reaching depths of approximately 3 km. Fig. 9C depicts proxy SAT data for Greenland and Antarctica.

    The blue and orange plots represent SAT at Antarctica's Dome C and Vostok respectively. Both datasets were given as temperature anomalies from current conditions, so we have shifted the plots to match with the present mean temperature of -55°C for this region of Antarctica (Petit et al., 1999). The orange plot employs the same principle as is used to estimate global

ice volume from foraminifera shells. Since precipitation containing the lighter $^{16}$O is more likely to evaporate, we expect to see greater proportions of $^{18}$O in the preserved ice when temperatures were higher. Through experiment, Dansgaard found this relationship to be approximately $T = 1.45\delta + 19.7$, where $\delta$ is the $\delta^{18}$O ratio (Dansgaard, 1964). The blue plot also uses this principle, but instead measures the ratio of hydrogen isotopes in the ice. Although the isotopic ratios of hydrogen and oxygen are closely related, they can deviate slightly due to the conditions in the region of ocean from which the precipitation

originated (Jouzel, 2013). Although different isotopes were measured in the two separate cores, we see a close agreement between the datasets.

    The Greenland SAT proxies are plotted in green and red, based on cores from the GRIP and NGRIP sites respectively. The green plot uses $\delta^{18}$O as a proxy for SAT and has been converted to temperature using $T = -0.1925\delta^2 - 11.88\delta - 211.4$ (Johnsen et al., 1995). This model was fitted to ice core data from GRIP so is likely to be more accurate than the generalised linear model.

This core has a $\delta^{18}$O record up to 250 kya. However, due to ice folding close to the bedrock, only the most recent 100 kyr of the record is reliable (members, 2004). We have shown the full record here because we only wish to compare it to our solution qualitatively. The red plot spans only 120 kyr but is considered reliable for this duration as it is from the separate drill site, NGRIP (members, 2004). The plot is also based on $\delta^{18}$O but additionally uses the nitrogen isotope ratio from the air bubbles trapped within the ice. Whilst air can still move around the compacting snow, the $^{15}$N isotopes are more likely to sink than

the lighter $^{14}$N. This enrichment of the lower portion of compacting snow is inversely related to the surrounding temperature, providing additional data for the SAT estimate (Kindler et al., 2014).



Notably, the data from Greenland provides a higher time resolution but significantly shorter time-span. This is due to the accumulation rates around the drill sites (Jouzel, 2013). With greater precipitation, there are more data available for the same duration. However, this reduces the time span captured by the same depth of ice.

The increased resolution in Greenland's ice core allowed for the discovery of the high frequency climate fluctuations known as Dansgaard–Oeschger events (Dansgaard et al., 1982; Johnsen et al., 1992). The cause of these fluctuations is not fully understood, though they appear to be local to the northern hemisphere and are believed to relate to changes in the Atlantic ocean due to freshwater perturbations (Dokken et al., 2013). These oscillations occur with a period of approximately 1.5 kyr (Grootes and Stuiver, 1997). Since our model is a linear combination of the significantly slower orbital parameters, these

fluctuations will not appear in our solution for SAT. This is the intended behaviour of our model since the phenomenon is intrinsic to Earth. Instead of reproducing the SAT data exactly, we wish to show that the SAT dynamics that result directly from the orbital variations are sufficient to explain the majority of the ice volume data.

Since our data only represents the most extreme of environments, we are careful not to overgeneralise to global SAT. However, it is worth noting that $S(t)$ might better reflect polar SAT as it holds the most significant influence on ice volume. We

therefore expect to find a reasonable qualitative agreement between the polar SAT data and the $S(t)$ solution that best reproduces the ice volume data.

## 4.5 Fitting

To estimate each physical parameter, we use the fit $p_i$ values from our phenomenological model along with 4 additional constraints. These relate to the range, and average, of bulk ocean temperature and SAT. We see BWT varies between -1°C and

2°C globally, whereas the mean SWT varies dependent on location, but has an approximate range of 5°C. Assuming these both contribute equally to the bulk ocean temperature gives an estimated bulk ocean temperature range of $\Delta_O = 4$°C. The average of the SWT data is approximately 19.4°C whilst the averaged BWT is 0.2°C, giving an estimated average of $\mu_O = 9.8$°C.

For the SAT data, we are restricted by the extreme locations of its sources, namely Greenland and Antarctica. We assume that these reflect the qualitative dynamics of global SAT over time but may have a differing range to the global signal, and will

certainly have a lower mean than the global average. We will therefore employ further sources to determine the $S(t)$ equation coefficients.

Thomas estimates that SAT increased $5.8 \pm 1.4$°C from the last glacial maximum to pre-industrial time (Schneider von Deimling et al., 2006), which was approximately 13°C (Lindsey and Dahlman, 2020). Since these periods span close to the minimum and maximum temperatures of the past 800 kyr, we use this to constrain the SAT between 7.2 and 13°C, giving an

estimated range of $\Delta_S = 5.8$°C and an estimated mean SAT of $\mu_S = 10.1$°C. We now have the necessary constraints to extract



**Table 3.** Parameter values for our physical model.

| Parameter | Value | Units |
|:---:|:---:|:---:|
| $a$ | $2.2 \times 10^7$ | $\mathrm{km}^3/°\mathrm{C}$ |
| $b$ | $2.8 \times 10^7$ | $\mathrm{km}^3/°\mathrm{C}$ |
| $c$ | 85 | $°\mathrm{C}$ |
| $d$ | 70 | $°\mathrm{C}$ |
| $e$ | 56 | $°\mathrm{C}$ |
| $f$ | 0.69 | $°\mathrm{C}$ |
| $\alpha_I$ | $1.4 \times 10^8$ | $\mathrm{km}^3$ |
| $\alpha_O$ | 7.5 | $°\mathrm{C}$ |
| $\alpha_S$ | $-14$ | $°\mathrm{C}$ |
| $\tau$ | 15 | kyr |

each physical parameter from the phenomenological model coefficients, where the equations to solve are

$$ac = p_1, \tag{28}$$

$$bd = p_2, \tag{29}$$

$$be = p_3, \tag{30}$$

$$bf = p_4, \tag{31}$$

$$a\alpha_O - b\alpha_S + \alpha_I = p_5, \tag{32}$$

$$\mathrm{Mean}[O(t)] = \mu_{\mathrm{O}}, \tag{33}$$

$$\mathrm{Range}[O(t)] = \Delta_{\mathrm{O}}, \tag{34}$$

$$\mathrm{Mean}[S(t)] = \mu_{\mathrm{S}}, \tag{35}$$

$$\mathrm{Range}[S(t)] = \Delta_{\mathrm{S}}, \tag{36}$$

where

$$\mathrm{Mean}[X(t)] = \frac{1}{800} \int_0^{800} X(t)\mathrm{d}t, \tag{37}$$

$$\mathrm{Range}[X(t)] = \max[X(t)] - \min[X(t)], \tag{38}$$

$$O(t) = c\zeta_\tau[\varepsilon(t)] + \alpha_O - \alpha_O e^{-t/\tau} + O_0 e^{-t/\tau}, \tag{39}$$

$$S(t) = d\varepsilon(t) + e\beta(t) + f\cos(\rho(t)) + \alpha_S. \tag{40}$$

The solution to these equations produces the physical model parameters shown in Table 3.



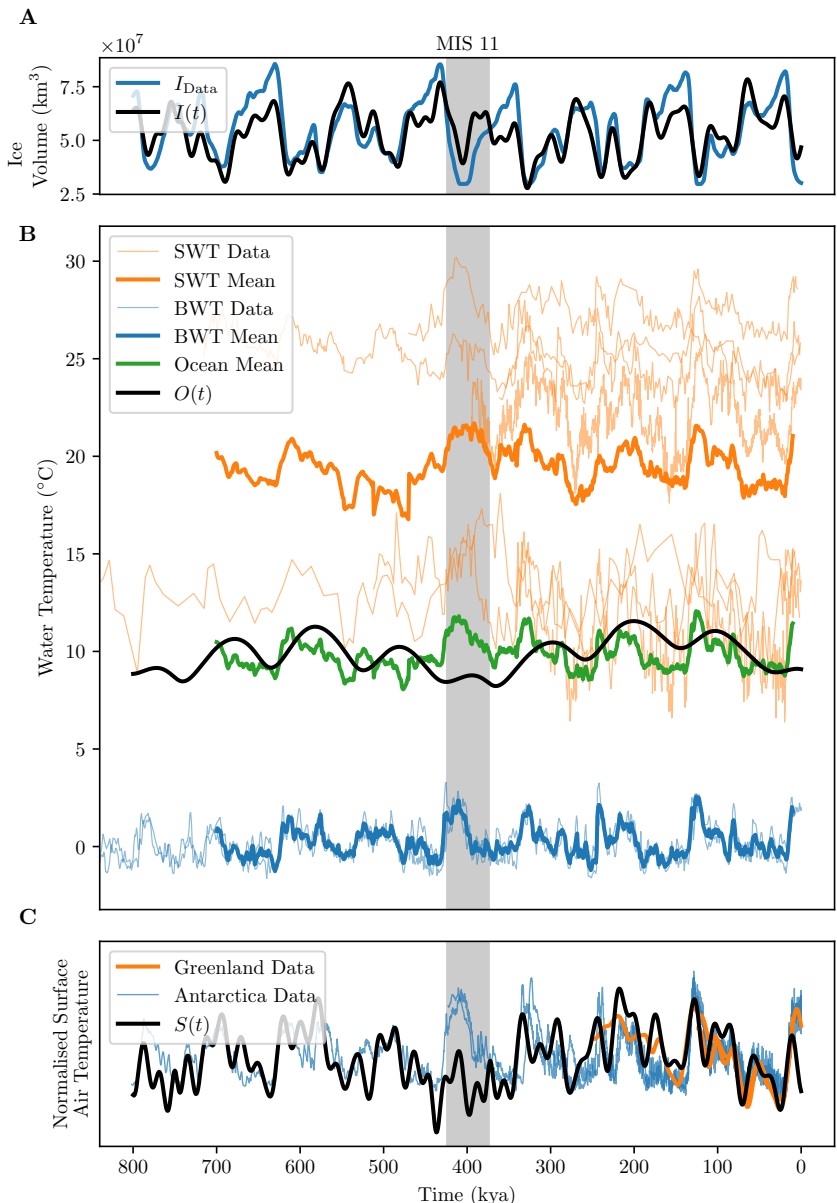

**Figure 10. A**: Our modelled ice volume compared with observational constraints (for details, see Appendix B). **B**: Our modelled bulk ocean temperature compared with the SWT and BWT data shown in Fig. 9. The mean SWT and BWT are averaged to give an approximate bulk ocean temperature (green) with the ocean solution $O(t)$ overlaid. To avoid over dependence on one data source, the averages are only calculated as far as 700 kya. **C**: Our modelled SAT compared with data from Fig. 9. The data only reflects glacial air temperatures so has been rescaled to align with the warmer surface solution $S(t)$ to show the qualitative similarity. We have also only included the smoothed signal for the Greenland data as it is too noisy to discern the large-scale dynamics. The MIS 11 interglacial period is highlighted in grey.



With these coefficients we can plot the proposed solutions for bulk ocean temperature $O(t)$ and SAT $S(t)$, the weighted sum of which gives the same solution for ice volume $I(t)$ shown in Fig. 4. These solutions are all shown together, alongside their relevant data, in Fig. 10.

In comparing the solutions for $O(t)$ and $S(t)$ with their respective proxy data, we see a similar agreement for the majority of the timespan, except around MIS 11 as with the ice volume solution from our phenomenological model. This artefact is discussed in Sec. 5. Aside from MIS 11, we see the $O(t)$ solution loosely matching the $100\,\mathrm{kyr}$ oscillations that appear in the averaged data. However, without the higher frequency obliquity and precession as inputs to $O(t)$, it is restricted to the frequencies present in eccentricity.

Investigating SAT data, it appears to reproduce the data well. Due to its faster response rate, and the inclusion of both obliquity and precession, a higher range of frequencies are captured. The quality of this fit is especially noteworthy because we have not directly fit to the SAT data. Instead, we used the ice volume data to determine the dynamics of $S(t)$, and then linearly scaled the solution using SAT data. This suggests that surface temperature can be well explained by the orbital parameters alone, and that it is a significant factor in determining ice volume. It is also worth noting that the solutions have similar ranges

when scaled by their respective coefficients in the ice volume equation. This suggests that their roles are of similar significance in governing the glacial-interglacial cycles.

    Now that we have determined the coefficients of our physical model, we can convert the simplified parameters in $O(t)$ to those used in the original ocean temperature model given by

$$\frac{\mathrm{d}O(t)}{\mathrm{d}t} = 9.591\gamma\varepsilon(t) + 353.8\gamma - mO(t) - n, \tag{41}$$

where $\gamma$, $m$, and $n$ are unknown.

    From fitting to the ice volume data, we attained values of $c = 85°\mathrm{C}$, $\alpha_O = 7.5°\mathrm{C}$, and $\tau = 15\,\mathrm{kyr}$. Using the substitutions outlined in Sec. 4.1, we convert back to get

$$\gamma = \frac{c}{9.591\tau} = 0.59, \tag{42}$$

$$m = \frac{1}{\tau} = 0.067\,\mathrm{kyr}^{-1}, \tag{43}$$

$$n = 353.8\gamma - \frac{\alpha_O}{\tau} = 210°\mathrm{C/kyr}. \tag{44}$$

Note that, although the constant heat loss term $n = 210°\mathrm{C/kyr}$ seems extreme, it is mostly cancelled out by the constant heat gain term $353.8\gamma$, resulting in a dynamic equilibrium. With these fitted parameter values, we can estimate the average heat loss rate of the ocean as

$$\mathrm{Mean}\,[mO(t) + n] = m\mu_O + n, \tag{45}$$

$$= 0.0670 \times 9.8 + 210, \tag{46}$$

$$= 211°\mathrm{C/kyr}, \tag{47}$$

where $\mu_O = 9.8°\mathrm{C}$ was estimated in Sec. 4.5.




It is encouraging that once fitting the free parameters of the ocean model, we attain a value of $\gamma = 0.59$. This means almost 60% of the extra insolation due to eccentricity is transferred as heat to the bulk ocean, which is physically valid. If the fitting

process had produced an optimal value of $\gamma > 1$, it would suggest that some form of amplification is required to produce the observed dynamics. By attaining a value of $\gamma = 0.59$, we have therefore shown that enough energy is present in the system for eccentricity to explain the range of ocean temperature without the need for an amplification term.

Additionally, the result of an almost constant heat loss rate lends support to our original simple calculation in Sec. 2, where we assumed a constant heat loss rate and $\gamma = 0.5$. Aside from these assumptions, we used measured values to estimate a

temperature rise of $6.4°C$ if maximum eccentricity was sustained for $20\,\text{kyr}$. This shows good agreement with the approximate $4°C$ rise expressed in the proxy data over the same period. Although eccentricity does not remain constant, this shows it can have a direct and significant impact on the ocean and cryosphere without the need for amplification.

A near constant heat loss rate is physically plausible due to the omission of the surface ocean layer when defining $O(t)$. The surface layer exchanges heat more easily with the atmosphere due to evaporation, whilst the deeper ocean is more insulated. It

is therefore plausible that an approximately $4°C$ change in bulk ocean temperature has little impact on its heat loss rate.

## 5 Discussion

We have presented a simple phenomenological model of global ice volume, assuming only a linear dependence on the orbital parameters. Aside from the interval around MIS 11, this model was able to reproduce the qualitative features of the ice volume data over the past $800\,\text{kyr}$. We then proposed a physical interpretation of the model, whereby intermediate variables repre-

senting ocean temperature and SAT respond to the orbitally governed insolation. The weighted sum of these two intermediate variables results in the same ice volume solution as before. The fit coefficients were physically plausible and gave solutions that qualitatively matched the ocean temperature and SAT data, apart from around MIS 11. This supports the hypothesis that the majority of ice volume dynamics can be explained by a linear model that is only driven by the orbital parameters.

### 5.1 Model limitations

For our physical model, we proposed variables $O(t)$ and $S(t)$ to represent the slow and fast parts of the ice volume equation respectively. We emphasise that, although these have been attributed to ocean temperature and SAT, they may also incorporate other geophysical mechanisms. Examples of these are the $CO_2$ cycle, which is connected to ocean dynamics, and direct glacial radiation, which would be an instantaneous function of insolation, similar to SAT. It is also possible that, instead of modelling the global averages for each variable, we are representing certain regions or internal components of them.

An example of this is a region known as the North Atlantic Deep Water (NADW), which is part of the Atlantic Meridional Overturning Circulation (AMOC). This circulation is characterised by warmer surface water that flows north towards the NADW, where it cools, becoming denser and sinking due to convection before flowing south. The NADW plays an important role in the oxygenation of Earth's deep oceans and the modulation of SAT through the loss of evaporative heat (Raymo et al., 1990). As a result, it has been suggested that the NADW is a significant factor in long-term climate variability (Rahmstorf,



2002). This more abstract interpretation of $O(t)$ could also lead to a better justification for the positive relationship between $O(t)$ and ice volume apparent in our physical model. Another issue with $O(t)$ representing global ocean temperature relates to the 400 kyr period that appears in its solution. From spectral analysis of the available ocean data, there is little evidence for a 400 kyr period. This could be due to the inaccurate and noisy nature of the data, though it is also possible that more complex dynamics are influencing ocean temperature, effectively removing the 400 kyr period from the data.

Although our ice volume solution largely matches the local trends of the data, it does not always match the magnitude. We present two possible explanations for this. One possibility relates to the inherent uncertainty in the data. As discussed in Appendix B, our chosen ice volume data has been aggregated from a number of different sources, and the BWT component of this signal was then removed using a model. Since the data are inherently uncertain, we would not expect the model to fit it perfectly. A second possibility is that Earth system feedbacks play a more significant role on ice volume at extreme

values. For example, the feedback that results from ice albedo can lead to accelerated ice growth or retreat, depending on the direction of the change. For moderate values of ice volume, any feedbacks were shown to be well approximated by our linear phenomenological model. However, it appears that the non-linear response to the orbital cycles becomes more pronounced around the highest 25%, and the lowest 12%, of ice volume. As a result, the linear model fails to match the amplitudes at these extrema in the data.

One interval where all variables fail to match even the local trend of the data is around MIS 11. This region is both the warmest, and the longest, interglacial over the past 800 kyr, which could be due in part to ice sheet instability. Hearty examined marine deposits in Bermuda and the Bahamas and proposed that a sea level rise of approximately 20 m above present took place during MIS 11 (Hearty et al., 1999). Hearty suggests that a 20 m sea level rise could only be achieved by the complete melting of Greenland and West Antarctica, as well as the partial melting of the far larger East Antarctica ice sheet. This is

substantiated by Christ, who uses the subglacial sediment in an ice core from northwestern Greenland to conclude that this region was ice free during MIS 11 (Christ et al., 2023).

Ice sheets such as West Antarctica are able to flow quickly through ice-streams, resulting in non-linear flow that is hard to model. Hollin points out that because these ice sheets are unstable, they can succumb to surges of melting with runaway effects (Hollin, 1965). Since the dynamics of these ice sheets are not well understood, we cannot explain why this occurred so

significantly in MIS 11 in particular. However, if this was the result of an Earth system runaway feedback, we would not expect it to appear in our orbitally forced solution of ice volume. We would also expect the surge in melting ice volume to result in a reduction in global albedo and an increase in atmospheric $CO_2$ as it is released from the ice sheets. This would then explain why ocean temperature and SAT are significantly higher than our physical model predicts during this period.

Another potential factor for the misalignment around MIS 11 is the Mid-Brunhes Event, which occurred around the same

time as MIS 11 and marked a global climatic shift. The Mid-Brunhes Event is characterised by a shift to warmer interglacial periods, along with an increase in the variance of atmospheric $CO_2$ and $CH_4$, Antarctic temperature, and ocean temperature (Barth et al., 2018). Although the cause of this event is not well understood, some suggest that it is due to a complex climatic response to magnitude changes in the orbital parameters (Jansen et al., 1986; Yin, 2013). Because MIS 11 likely resulted from either a



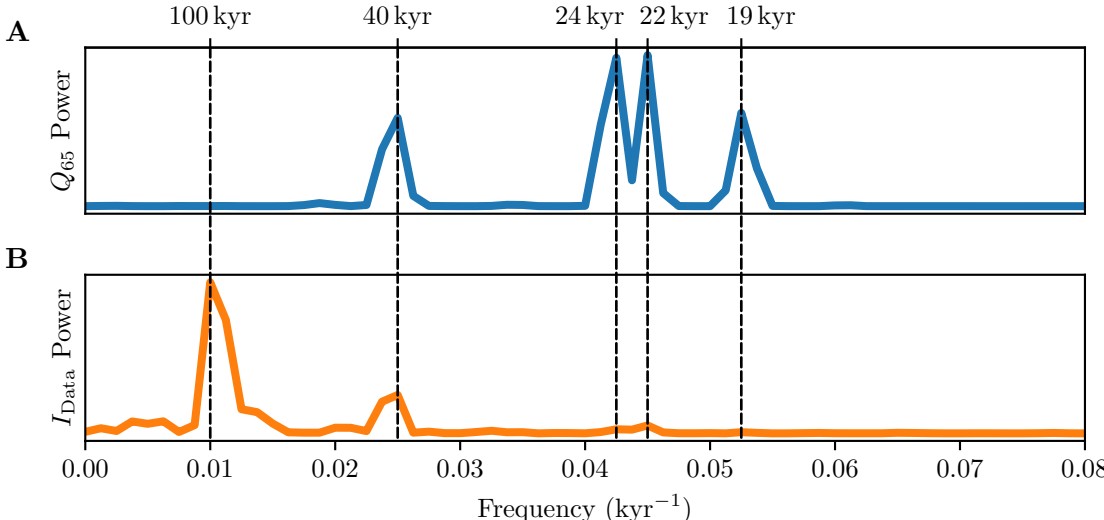

**Figure 11.** Power spectra for $Q_{65}$ (**A**) and the ice volume data (**B**) for the past 800 kyr. Periods corresponding to peak frequencies are marked by dashed lines.

non-linear response to orbital forcing, or from an Earth system process, we would not expect our linear OFPA model to capture
them.

The purpose of our model was to isolate the effect of orbital forcing on the Earth system, and therefore it cannot produce unforced oscillations. We recognise that intrinsic cycles on Earth likely play a role in the ice volume dynamics, however our model demonstrates that they are not required to explain the majority of the ice volume dynamics.

## 5.2 $Q_{65}$

One of the main aims for this paper was to address the hypothesis that eccentricity's impact on insolation is too small to drive the glacial-interglacial cycles without amplification. Justification for this hypothesis often leans on the $Q_{65}$ insolation measure, which is commonly used in this area (Paillard and Parrenin, 2004; Imbrie et al., 1993; Birchfield and Ghil, 1993a). As shown in Fig. 11, the 100 kyr period relating to eccentricity is imperceptible in the $Q_{65}$ power spectrum compared to the frequencies relating to obliquity and precession. It is therefore logical to hypothesise that one or more feedback mechanisms are amplifying
eccentricity's impact to produce the significant 100 kyr peak shown in the ice volume power spectrum.

Since $Q_{65}$ measures the insolation at a single latitude on a single day each year, we question how well it can represent the impact of each orbital parameter on the global insolation profile. By instead evaluating eccentricity's effect on Earth's average annual insolation, we have shown that it is capable of producing significant changes to the global ocean temperature without amplification.





### 5.3 Physical Variables

In Fig. 10, we see that $O(t)$ only weakly aligns with the averaged ocean temperature data. This is understandable, since the purpose of these physical variables is not to accurately reproduce their corresponding data. Rather, their purpose is to depict the component of each mechanism's dynamics that arise due to orbital forcing. Since our modelled ice volume depends linearly on these two variables, we hope to have isolated the ice volume dynamics that result from a linear response to orbital forcing.

This approach would not work if most of ocean temperature or SAT dynamics were intrinsic or non-linear. However, we have shown that the SAT data can be well reproduced using only the orbital parameters as inputs. The ocean temperature data are less accurately reproduced through this method, though we were able to demonstrate that eccentricity alone could explain the magnitude of change seen in the data. This demonstration used measured values and the free parameter $\gamma$, which was estimated to be $0.59$ from fitting to the ice volume data. By attaining a fit value of $0 \leq \gamma \leq 1$, the ocean temperature model remains physically feasible, and can therefore lend support to the OFPA school of thought.

### 5.4 Sensitivity Analysis

In order to determine the overall sensitivity of the ice volume solution on the parameters in our physical model, we perturb them randomly and show the resultant ice volume solution. The results of this can be seen in Fig. 12. Here we have rescaled each of the parameters by a different value drawn from the normal distribution $\mathcal{N}(1, \sigma^2)$. The figure shows 15 iterations of this process for both $\sigma = 0.1$ and $\sigma = 0.2$. In order to maintain the correct offset, the constant term $\alpha_I$ was determined as a function of the other parameters.

In the $\sigma = 0.1$ case, we see close agreement with the optimal fit in all 15 iterations. Only when the perturbations are drawn from $\mathcal{N}(1, 0.2^2)$ do we see a significant deviation from the optimal fit, though the qualitative behaviour is still preserved. This suggests that the model is not highly sensitive to any of the parameters, reducing the risk of overfitting. Furthermore, it suggests that in the absence of anthropogenic forcing or rare climatic shifts, the system is fairly predictable.

### 6 Conclusions

Here we have shown that since the MPT, global ice volume dynamics can be approximated by a linear function of the orbital parameters. Our results show that the data may not necessitate switching mechanisms nor unforced oscillations, supporting the Orbital Forcing with Potential Amplification approach.

Additionally, through estimating the annually averaged insolation reaching the bulk ocean, we found that eccentricity is capable of producing significant changes to the ocean and cryosphere without amplification. We propose this important effect may have previously been overlooked due to the common use of $Q_{65}$, which we find to be insufficient in representing the impact of each orbital parameter on the global insolation profile.

Our phenomenological model also provides a mechanism that is able to remove eccentricity's $400\,\mathrm{kyr}$ period from the ice volume solution, in line with the data. For our physical model, we propose this mechanism relates to the bulk ocean temperature,





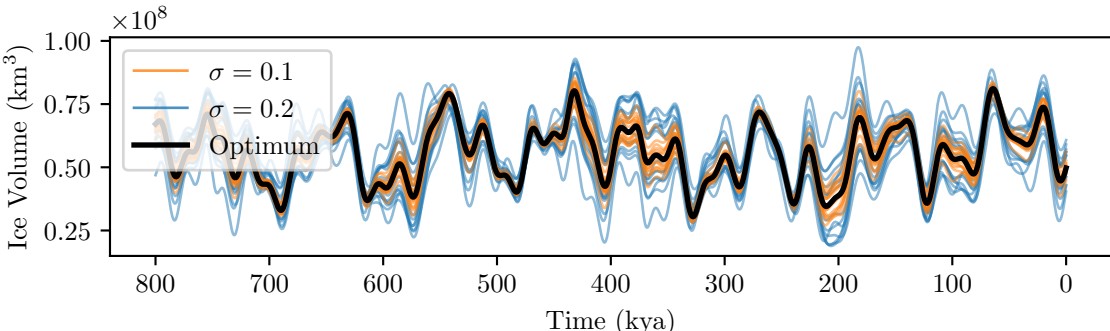

**Figure 12.** Set of solutions for random parameter perturbations. All parameters in the physical model have been perturbed each iteration accept for $\alpha_I$, which was chosen as a function of the other parameters each time to maintain the correct offset. The parameters are rescaled by coefficients chosen from $\mathcal{N}(1, \sigma^2)$ with simulations run for $\sigma = 0.1, 0.2$.

which responds to eccentricity on a slower timescale than the SAT. By taking the difference of these two variables in our ice volume equation, we produce a signal that resembles the change in eccentricity. This signal has significantly less power in the 400 kyr band which results in a far better fit to the ice volume data. To emphasise the significance of this mechanism, we systematically refit the model with each subset of the terms in the phenomenological model. We found the pair of lagged and

instantaneous eccentricity terms to be the greatest contributor to the model's accuracy, suggesting that ice volume has more dependence on the change in eccentricity that its absolute value.

Although we are modelling this Earth system as linear, we acknowledge that non-linear dynamics are at play, and are more important at certain times. Meltwater erosion and ice-albedo are two such feedbacks that could lead to the accelerated change in ice volume expressed in the data. However, since the MPT, there have only been eight 100 kyr glacial-interglacial cycles. If

we are to unearth the mechanisms that govern this relatively brief period of ice volume dynamics, there is value in imposing as few assumptions as possible. Since the data are adequately reproduced by our model, we propose that ice volume dynamics predominantly follow an approximately linear response to the orbital parameters.

*Code and data availability.* https://github.com/liamwheen/Wheen_23

## Appendix A: Orbital Data

For all models in this paper, we use the orbital solutions produced by Laskar in 2004 (Laskar et al., 2004). To estimate the reliability of these solutions over the past 1 myr, we can compare them to the previous solutions produced by Laskar in 1993 (Laskar et al., 1993), as well as the solutions produced by Berger in 1999 (Berger and Loutre, 1999). Laskar's 1993 solution for eccentricity deviates from the more recent equivalent by an average of 0.4% over the past 1 myr, whilst Berger's



eccentricity solution deviates by an average of 1.2%. The obliquity and precession deviations are significantly less in both

cases. The differences between these solutions suggests an uncertainty on the order of 1%. This is an acceptable level for our purposes due to the less accurate nature of the proxy data we rely on to estimate global ice volume. To justify this, we tried fitting our model with all three orbital solutions and found that the difference was negligible when comparing to the ice volume data.

**Appendix B: Ice Volume Data**

Ice volume proxy data is commonly derived by analysing isotopic concentrations in drill cores obtained from continental ice sheets or the ocean floor. For oceanic records, the $\delta^{18}$O value is often used. This represents the ratio of $^{18}$O to $^{16}$O found in foraminifera shells deposited on the ocean floor. The ratio provides insights into the proportion of $^{16}$O, which evaporates more easily due to an effect known as Rayleigh distillation, stored within glacial ice during a specific period. Many datasets have been produced through this method, however there is a lack of agreement between them due to the complexity of the

reconstruction process. By aggregating multiple datasets, we can expect the impact of anomalies and location specific behaviour to be reduced. Lisiecki and Raymo collected $\delta^{18}$O data from 57 marine sediment cores around the world and combined them using an automated graphic correlation algorithm (Lisiecki and Raymo, 2005). Although $\delta^{18}$O is related to the volume of glacial ice, it is also affected by the local bottom water temperature (BWT) in which the foraminifera lived. The ambient temperature of the foraminifera changes the amount of $^{18}$O that is absorbed into their shells (Epstein et al., 1953). With only

one measure to represent two quantities, researchers have used the ratio of Magnesium to Calcium in the shells to estimate BWT. Despite this, the Mg/Ca ratio within the foraminifera shells can also vary due to calcitep dissolution, the degree of which increases with depth (Rosenthal et al., 2000).

Instead of relying on additional measures, Bintanja uses the same $\delta^{18}$O stack produced by Lisiecki and Raymo to create an internally consistent model of mean surface air temperature, linked with ice volume and global sea level (Bintanja et al., 2005).

This then produces time series for the ice volume and BWT contributions to the $\delta^{18}$O data. The outputs of this model were compared against other independent proxy datasets and showed close agreement over the full timespan. We therefore opt to use Bintanja's modelled ice volume data to fit our coefficients.

The ice volume data from Bintanja's model is expressed as a proportion of the total benthic $\delta^{18}$O value, measured in parts-per-thousand (‰). In order to evaluate our model by comparing to other proxy data, we convert the ice proportion of $\delta^{18}$O into

ice volume. For this, we assume a linear relationship between ice volume and the contribution to the $\delta^{18}$O data. We then use estimates for the physical values needed to linearly transform the ice contribution into ice volume.

The current estimates of ice volume by location are approximately $2.7 \times 10^7 \, \text{km}^3$ in Antarctica (Fretwell et al., 2013), $2.99 \times 10^6 \, \text{km}^3$ in Greenland (Morlighem et al., 2017), and $1.58 \times 10^5 \, \text{km}^3$ in all other regions (Farinotti et al., 2019). This gives an approximate total of $3.0 \times 10^7 \, \text{km}^3$. Lambeck estimates that the ice volume during the last glacial maximum was ap-

proximately $5.2 \times 10^7 \, \text{km}^3$ greater than at present (Lambeck et al., 2014). This occurred around 21 kya, giving us an anchor point to scale the range of $\delta^{18}$O to ice volume. We can now convert the ice volume component of the benthic $\delta^{18}$O data $I_{\text{Benth}}$



into total ice volume $I_{\mathrm{Data}}$ using

$$I_{\mathrm{Data}} = m I_{\mathrm{Benth}} + c, \tag{B1}$$

where

$$m = \frac{I_{\mathrm{Data}}(0) - I_{\mathrm{Data}}(-21)}{I_{\mathrm{Benth}}(0) - I_{\mathrm{Benth}}(-21)} \approx \frac{3.0 \times 10^7 - 8.2 \times 10^7}{0.0 - 1.0} = 5.2 \times 10^7 \,\mathrm{km}^3/\text{\textperthousand}, \tag{B2}$$

$$c = I_{\mathrm{Data}}(0) - m I_{\mathrm{Benth}}(0) \approx 3.0 \times 10^7 - 5.2 \times 10^7 \times 0.0 = 3.0 \times 10^7 \,\mathrm{km}^3. \tag{B3}$$

*Author contributions.* Liam Wheen performed the research and wrote the paper.

Oscar Benjamin advised the research and wrote the paper.

Cameron Hall advised the research and wrote the paper.

Jerry Wright advised the research and wrote the paper.

Tom Gernon wrote the paper.

*Competing interests.* There are no competing interests.

*Acknowledgements.* L.W. gratefully acknowledges the support of the Engineering and Physical Sciences Research Council.

T.G. gratefully acknowledges the support of the WoodNext Foundation, a component fund administered by the Greater Houston Community Foundation.



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
