# Peer review of "The Largely Linear Response of Earth's Ice Volume to Orbital Forcing"

_EGUsphere, 2023_

## Referee Comment (RC1)

**Report on "The Largely Linear Response of Earths Ice Volume to Orbital Forcing"**

**1 There are problems with providing plausible mechanisms behind the linear equations presented here, especially the negative coefficient of eccentricity.**

A key component of the fit of this model is the removal of (most of) the 400K cycle, which is accomplished by effectively using the derivative of eccentricity. However, the mechanism by which the derivative of eccentricity plays such a key forcing role strains credibility. In this regard, let us examine the signs of the key parameters in the dynamical equations:

Note that each of the $p_i$ are positive by Table 2.

$p_2$ is prefaced by a minus sign in Eqn. 7, meaning that $I(t)$ (ice volume) is equilibrating towards $-\varepsilon(t)$ along with $+\tilde{\varepsilon}(t)$.[1]

This seems unphysical since it means that, while larger values of $\varepsilon$ properly have a warming effect (reducing the ice volume[2]), higher values of the ocean temperature $\tilde{\varepsilon}$ paradoxically have a cooling effect, increasing ice volume. This sign change goes to the heart of the method since, as the authors explain, the the model is driven by the derivative of eccentricity (along with precession and obliquity), and it is hard (at least for me) to envisage a plausible mechanism for $d\varepsilon/dt$ to play a significant role in ice volume dynamics.

This aspect of the model is manifested in Fig. 8 and Eqn. 19, which show surface and ocean temperatures having opposite effects on ice volume. As expected, warm surface temperatures reduce ice volume. But why would warm ocean temperatures raise ice volume? The authors propose that higher ocean temperatures would lead to an increase of precipitation and hence an increased buildup of ice sheets. However, if one looks at the ocean temperature data in Fig. 10, one finds a very strong alignment of peaks and valleys with the $I_{\text{Data}}$ curve[3] *once it has been flipped vertically for a sign change*. This is in accord with the general picture one sees for various temperature proxies around the world as moving approximately in concert with the ice volume, with relatively small lags compared with the timescales involved in ice sheet dynamics. So it is hard to accept the flipped ocean temperature as a driver in the model. More broadly, the simplicity and reasonableness of this linear model should be questioned, because of its wrong sign driver and its sequencing of surface and water temperature (resulting in compounded time lags).

**2 Inconsistency with Lisiecki's observations; What does nearly linear mean and is it a useful concept for ice dynamics?**

It is important to take account of Lisiecki's paper on the negative correlation between 100K power in climate with that of eccentricity (Nature Geoscience, 2010). This leads to the idea that low $\varepsilon$ dampens the precession effect, creating conditions for ice volume to continue building up. She writes: "I propose that the anticorrelation arises from the strong precession forcing associated with strong eccentricity forcing, which disrupts the internal climate feedbacks that drive the 100,000-yr glacial cycle. This supports the hypothesis that internally driven climate feedbacks are the source of the 100,000-yr climate variations." This observation has been important in subsequent works focusing on understanding the nonlinear phenomena leading to the dominance of 100K cycles, and to their continuation even during low eccentricity epochs such as Marine Isotope Stage (MIS) 11.

Since the present paper's linear model relies on direct eccentricity forcing, it is clear that there is a positive or neutral correlation of its 100K power with that of eccentricity. In the time period investigated (0-800kya), the main issue is to account for the unusually large amplitude of the glacial maximum and subsequent termination and long interglacial in MIS 11.

Is it a meaningful distinction to say the climate system is largely linear if it is linear except when it isn't? One should note that successful nonlinear models, such as models with bistability, are actually approximately linear when not making transitions between stable modes.

**3 MIS 11 should not be dismissed as an unimportant exception.**

Failure to explain the record around MIS 11 is an indication that a given model is missing a key component of Pleistocene climate dynamics. As I mentioned above, a system may be "linear most of the time" but the exceptional times may have profound effects, which call for a fundamentally nonlinear approach.

We know that the power spectrum of the ice volume record is dominated by the frequencies seen in the orbital parameters. This means that if one pulls those frequencies from the orbital parameters and combine them linearly while adjusting amplitudes and phases to match those of the ice volume record, one can get a pretty good match. The tricky part of the modeling problem is finding a physically reasonable set of equations that behaves as indicated in the record, amplifying some frequencies (100K) and suppressing others (400K), while maintaining the observed phase relationships. If one adopts eccentricity as a primary forcing variable, one can increase the 100K power – but then one has a problem with MIS 11. So even putting aside the question of whether eccentricity forcing is reasonable (which most authors have thought is not the case), one finds it necessary to work with nonlinear equations.
* * *
[1] Following the authors' notation, $\varepsilon$ is eccentricity, and $\tilde{\varepsilon}$ is a slow variable ($\sim$ ocean temperature), which resembles the eccentricity signal with a lag.

[2] This agrees with the historical association between high $\varepsilon$ and low ice volume.

[3] Ice volume inferred from scaled version of benthic $\delta^{18}O$, see (B1-B3)